# Identifying and Mitigating Errors in Gradient Aggregation of Distributed Data Parallel Training

## Abstract

Identifying and recovering from hardware failures is important in fault-tolerant distributed training to guarantee system efficiency. However, some hardware-related silent data corruption errors during gradient aggregation like bit corruptions or communication noise, are difficult to capture and address, leading to slow or failed convergence. To understand and mitigate these errors, we first mathematically formulate and generalize them as gradient inconsistency. Then, we theoretically analyze how it leads to model divergence accumulated during training and the failed convergence. Based on the analytical study, we design `PAFT`, a fault-tolerant distributed training system with dynamic and asynchronous parameter synchronization. `PAFT` includes two parts: (1) `PAFT-Sync`, which mitigates model divergence by periodically synchronizing parameters, and (2) `PAFT-Dyn`, which minimizes synchronization overhead through dynamic training overlap and synchronization frequency scheduling based on profiled error degrees. Together, they ensure efficient model convergence at scale. The fault-tolerant synchronization in `PAFT` is optimized to support commonly used optimizers, e.g., Stochastic Gradient Descent (SGD), SGD momentum, and Adam. We implement `PAFT` on PyTorch Distributed and train ResNet, GPT-2, and LLaMA-2 on $4 \sim 32$ GPUs. Experimental results show that `PAFT` efficiently defends against gradient aggregation error degrees while maintaining training performance.

## 1 Introduction

To efficiently train deep learning (DL) models (He et al., 2016) and large language models (LLMs) (Radford et al., 2018; Chung et al., 2022), high-performance and large-scale distributed training frameworks have been proposed (Rasley et al., 2020; Narayanan et al., 2021; 2019; Tang et al., 2023). Frequent system failures suspend training and require manual recovery from checkpoints, significantly reducing system efficiency and GPU utilization (up to 43%) (Maeng et al., 2021; Wang et al., 2023b). Approximately 178,000 GPU hours were wasted during the OPT-175B training (Zhang et al., 2022) due to various failures like MPI and CUDA errors (Humbatova et al., 2020), and hardware failures such as GPU malfunctions (Hu et al., 2024), electronic breakdowns, and node failures (Wang et al., 2023b; Hu et al., 2024). Many existing studies focus on improving the robustness and efficiency of the system through fast recovery (Wang et al., 2023b; 2024; Narayanan et al., 2021) or elastic training (Thorpe et al., 2022; Harlap et al.; He et al., 2023a).

However, unlike system failures, *silent data corruption (SDC) errors* (Wang et al., 2023a; Fiala et al., 2012; Bacon, 2022; He et al., 2023b), which do not directly interrupt training, are increasingly affecting model quality and convergence. As reported in LLaMA-3 pretraining cluster and Fire-Flyer cluster, SDC errors have become the main cause of LLM convergence issues, and the secondary cost of fault tolerance during pretraining (Dubey et al., 2024; An et al., 2024), harming the reliability and efficiency of GPU clusters at extensive scale. (We provide more real-world error types and frequency during LLM pretraining in Appendix D).

In this work, we consider the errors happen during gradient aggregation (GA), which are caused by hardware failures like bit corruptions (Jeon et al., 2019; Tiwari et al., 2015; Gao et al., 2023; Hu et al., 2024) and communication noise on network links (Hu et al., 2024; Gill et al., 2011; Tan et al., 2019;

Gao et al., 2023; Khan et al., 2023), as shown in Fig. 1. Specifically, the communicated messages are aggregated and broadcasted with noise, leading to different gradients on workers, which results in slow or failed convergence. To this end, we propose the following research questions.

*How do silent errors in gradient aggregation influence distributed training and how to capture and mitigate them?*

In this work, we formulate and generalize *gradient inconsistency* (in Section 2) errors, where workers obtain different noisy averaged gradients instead of the accurate averages. We then theoretically demonstrate that this gradient inconsistency leads to accumulated model divergence (in Section 3), resulting in failed convergence. Additionally, we quantify the convergence error theoretically concerning the degree of gradient inconsistency.

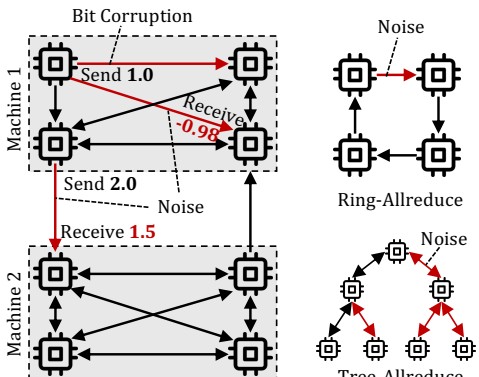

Figure 1: SDC errors lead to GA errors during distributed training. We provide more discussions about real-world cases in Appendix D.

To address the GA errors at scale, we design `PAFT`, a fault-tolerant distributed training system with two components: `PAFT-Sync` and `PAFT-Dyn`. `PAFT-Sync` periodically synchronizes model parameters with a frequency $H$ to eliminate the model divergence. Then, `PAFT-Dyn` overlaps synchronization with the training process through asynchronous communication to save parameter synchronization overhead. To further reduce unnecessary communication costs, `PAFT-Dyn` adjusts the synchronization frequency $H$ according to the signal-to-noise ratio as observed in our theoretical convergence analysis. Our theoretical and empirical studies show that `PAFT` can alleviate accumulated model divergence, ensuring training convergence.

We implement `PAFT` on PyTorch Distributed (Ansel et al., 2024) and extend it to DeepSpeed (Rasley et al., 2020) and Megatron (Narayanan et al., 2021) for real-world distributed LLM training deployment. We summarize our contributions as follows:

- We formulate gradient inconsistency caused by silent GA errors. We theoretically analyze how it leads to accumulated model divergence and failed convergence.

- We design `PAFT`, a fault-tolerant distributed training system to alleviate the gradient inconsistency. We theoretically prove that `PAFT-Sync` can illuminate the model divergence and ensure convergence. To reduce the extra communication overhead, we design `PAFT-Dyn` to overlap synchronization with training, and adjust the synchronization frequency with respect to the profiled error degree based on the theoretical analysis.

- We conduct real-world experiments with 8-node GPU cluster with $4 \sim 32$ GPUs to train ResNet-18 with CIFAR-10 (Krizhevsky et al., 2010), ResNet-50 with CIFAR-100 (Krizhevsky et al.), and LLMs including GPT-2 (Radford et al., 2019) and LLaMA-2 (Touvron et al., 2023) with OpenWebText (Gokaslan et al., 2019) and Alpaca (Taori et al., 2023). We consider noises with different patterns to simulate the SDC errors with different degrees. Results show that our method can successfully mitigate these errors.

## 2 PRELIMINARIES

We first present the preliminaries of distributed training, incorporating both image classification (He et al., 2016) and language modeling tasks (Radford et al., 2019). Then, we formulate the gradient inconsistency caused by the SDC errors during communication. With a model parameterized by $\theta \in \mathbb{R}^d$, and sampling data $x \sim \mathcal{D}$, the object function is usually defined as (Bottou et al., 2016)

$$\min_{\theta} F(\theta) \triangleq \mathbb{E}_{x \sim \mathcal{D}} f(\theta; x), \tag{1}$$

in which the specific definition of $f(\theta; x)$ depends on the task, and it is a general formulation in many deep learning optimization problems (Dean et al., 2012). For image classification, the $f(\theta; x) = l(\rho_\theta(x_i), x_o)$, where $x_i$ is the data inputs, $x_o$ the labels in the data sample, $x = (x_i, x_o)$,

$\rho_\theta(x_i)$ is the output of model $\rho_\theta$, $l$ is any classification loss function, like the cross-entropy. For next-word prediction in LLMs (Radford et al., 2019; Yang et al., 2019), the $f(\theta; x) = l(\rho_\theta(x_{1:n}), x_{n+1:N})$, where the sequence length of the $x$ is $N$. Given the seen tokens indexed by $1 : n$, the model predicts the unseen tokens indexed by $n + 1 : N$.

**Distributed SGD (DSGD).** In distributed training, multiple workers $\mathcal{M} = \{m | m = 1, 2, ..., M\}$ collaboratively optimize $\theta$. In $t$-th iteration, each worker calculates the local gradient $g_m(\theta_t^m)$. Then, the training system uses collective communication (Shi et al., 2021a; Thakur et al., 2005; Tang et al., 2020) or a parameter server (Jiang et al., 2020; Tang et al., 2020) to aggregate and broadcast the averaged gradient across workers to update model parameters $\theta$. This distributed gradient computation and model updating can be formulated as follows.

$$\bar{g}_t = \frac{1}{M} \sum_{m \in \mathcal{M}} g_t^m(\theta_t^m; x_t^m) = \frac{1}{M} \sum_{m \in \mathcal{M}} \nabla f(\theta_t^m; x_t^m), \ x_t^m \sim \mathcal{D}_m, \tag{2}$$

$$\theta_{t+1}^m = \theta_t^m - \eta_t \bar{g}_t, \tag{3}$$

where $\mathcal{D}_m$ represents dataset on worker $m$, $g_t^m(\theta_t^m; x_t^m)$ represents the local gradient of $f(\theta_t^m)$ of worker $m$ at iteration $t$, and the $\theta_t^m$ is updated with the average of local gradients $\bar{g}_t$. Normally, local dataset $\mathcal{D}_m$ has the same distribution as $\mathcal{D}$ in distributed training. We write $g_t^m(\theta_t^m; x_t^m)$ as $g_t^m$ for simplicity. Note that all models are initialized as $\theta_0$, and all workers utilize the same averaged gradient $\bar{g}_t$ to update their local models. Thus, there is $\theta_t^m = \theta_t$ during the training process.

**Errors in Distributed Averaging Gradients.** The SDC errors (Hu et al., 2024; Gao et al., 2023) in distributed training (Malcolm, 1971; Saad, 2020) actually add the noise on the estimated average gradient $\bar{g}_t$. Thus, workers finally obtain different noised gradients $\tilde{g}_t^m$ as follows.

**Definition 2.1.** (*Inconsistent Gradient*). The noised averaged gradient $\tilde{g}_t^m$ is called inconsistent gradient, if there is an individual noise $\epsilon_t^m$ generated depending on $m$-th worker added on $\bar{g}_t$.

$$\tilde{g}_t^m = \bar{g}_t + \epsilon_t^m, \ \epsilon_t^m \sim \mathcal{N}(0, \sigma^2), \tag{4}$$

in which noise $\epsilon_t^m$ is sampled from a Gaussian distribution $\mathcal{N}$ with mean of 0 and variance of $\sigma^2$.

**Noise Degree and Patterns.** The small $\sigma^2$ can represent the small communication noise and less frequent SDC happening. On the contrary, the large $\sigma^2$ can represent the larger noise like bit corruptions (Jeon et al., 2019; Hu et al., 2024) and more frequent happening. We consider both of these two patterns in our experiments. The noises may not consistently follow a consistent pattern during training. We also consider the burst pattern of large noise (like bit corruption) that accidentally happen during training in experiments (Section 5).

## 3 ANALYSIS OF THE FAILED CONVERGENCE

Fig. 2(a) shows training ResNet-18 with CIFAR-10 dataset across 4 workers with and without noises $\epsilon_t^m$ with different $\sigma^2$ ranging from $0.0001 \sim 1.0$. Results show that even the small noise 0.001 also leads to failed training convergence.

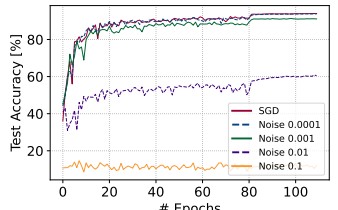 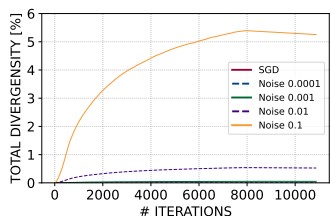

(a) Convergence Gap      (b) Model Divergence

Figure 2: Training ResNet-18 with gradient inconsistency on 4 workers.

### 3.1 ACCUMULATED MODEL DIVERGENCE

To understand and address this problem, we theoretically and empirically show how the gradient inconsistency (Eq. 4) leads to failed convergence. With the noised averaged gradient, the model updating process becomes from Eq. 3 as:

$$\theta_{t+1}^m = \theta_t^m - \eta_t \tilde{g}_t^m = \theta_t^m - \eta_t \bar{g}_t - \eta_t \epsilon_t^m. \tag{5}$$

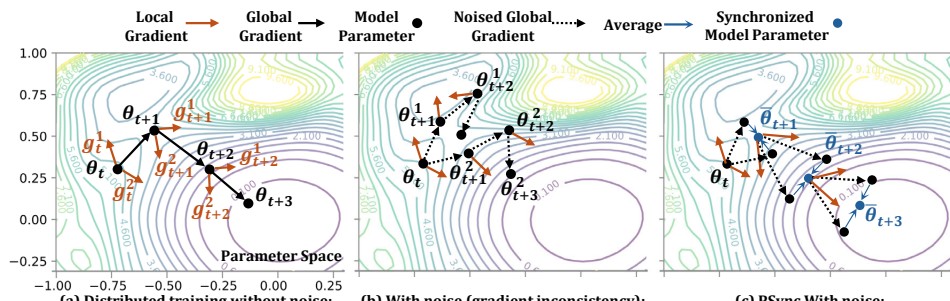

Figure 3: The trajectory of model parameters with training with two workers with/without noise and training with `PAFT`.

At $t$-th iteration, local models $\{\theta_t^m | m \in \mathcal{M}\}$ are updated towards different directions $\tilde{g}_t^m$. Thus, this leads to diverged model parameters $\theta_t^i \neq \theta_t^j \neq \theta_t$, instead of the same $\theta_t$ in normal DSGD (Eq. 3). With training goes on, models $\theta_t^m$ gradually diverge from each other. We define the averaged model $\bar{\theta}_t = \frac{1}{M} \sum_{i=1}^M \theta_t^i$ and model divergence $\Delta_t^m = ||\bar{\theta}_{t+1} - \theta_{t+1}^m||$ to measure it. Fig. 2(b) shows the empirical accumulated model divergence during training. Larger noise (higher $\sigma^2$) introduces more divergence. This aligns with training convergence curves in Fig. 2(a), where larger $\sigma^2$ leads to a larger accuracy drop or failed convergence. Fig. 3 (b) illustrates how the noised gradients interupt convergence. Specifically, the gradients obtained on a local model $\theta^1$ result in biased gradient direction of model $\theta^2$. And local models are optimized with larger divergence.

**Lemma 3.1** (Increasing Model Divergence). *With the same initial point $\theta_0^m = \theta_0$ across workers $\{m | m = 1, 2, ..., M\}$, DSGD with noise $\epsilon_t^m \sim \mathcal{N}(0, \sigma^2)$ introduces accumulated model divergence $\Delta_t^m$ during training:*

$$\mathbb{E}||\bar{\theta}_{t+1} - \theta_{t+1}^m||^2 = \frac{(M+1)\sigma^2}{M} \sum_{s=0}^t \eta_s^2. \tag{6}$$

**Remark.** Lemma 3.1 shows that the divergence $\Delta_t^m$ will be accumulated with the noise during training. This may lead to meaningless gradient estimation. Specifically, if the model $\theta_t^1$ is far away from the other model $\theta_t^2$, the gradient $\nabla f(\theta_t^1; x)$ has no useful descent information about the $\theta_t^1$ in the parameter space.

### 3.2 Convergence Analysis of Noised DSGD

**Assumption 3.1.** The following assumptions are commonly used in deep learning (Bottou et al., 2016): (1) Bounded variance: $\mathbb{E}_m ||g^m(\theta) - \nabla F^m(\theta)||^2 \leq \sigma_g^2$; (2) Bounded gradient magnitude: $\mathbb{E}_m ||g_m^m(\theta)||^2 \leq G^2$. The $\nabla F^m(\theta) = \mathbb{E}_i\, g^m(\theta)$ and $\nabla F(\theta) = 1/M \sum_{m \in \mathcal{M}} \nabla F^m(\theta)$, and the bounded variance comes from sampling bias of the dataset on worker $m$.

Now, we have the following theorem to show that it is difficult to tune the learning rate to have a good convergence speed.

**Theorem 3.2.** *(Convergence with noised training.) With object function defined in Eq. 1 satisfying Assumption 3.1, DSGD with noise $\epsilon_t^m \sim \mathcal{N}(0, \sigma^2)$ has the following convergence bound*

$$\frac{1}{T} \sum_{t=0}^{T-1} \eta_t \mathbb{E}(f(\bar{\theta}_t) - f^*) \leq \underbrace{\frac{2\mathbb{E}||\bar{\theta}_0 - \theta^*||^2}{T}}_{T_1} + \underbrace{\frac{2(\sigma_g^2 + \sigma^2)}{TM} \sum_{t=0}^{T-1} \eta_t^2}_{T_2}$$

$$+ \underbrace{\frac{4L\sigma^2(M+1)}{TM} \sum_{t=0}^{T-1} \eta_t \sum_{s=0}^{t-1} \eta_s^2}_{T_3}.$$

**Remark.** In Theorem 3.2, $T_1$, $T_2$ converge with respect to training iteration $T \to \infty$, $T_3$ only converges when setting $\eta_t = 0$. However, the zero learning rate does not have any practical effect on decreasing the object function. To alleviate the model divergence in Lemma 3.1 and $T_3$ in Theorem 3.2, we propose `PAFT` in Section 4.

## 4 PERIODICAL PARAMETER SYNCHRONIZATION

As discussed in Section 3, the root cause of the failed convergence is the optimization of local model parameters in different directions. In this section, we begin with a straightforward but systematic solution to this issue, parameter synchronization (Section 4.1). To minimize the additional overhead, we designed `PAFT-Sync` to efficiently ensure training convergence (Section 4.2 and 4.3).

---

**Algorithm 1** Distributed training with `PAFT-Sync`

---
**Input:** Initialized model $\theta_0$, dataset $\mathcal{D}$, workers $\mathcal{M}$, total iteration $T$, learning rate $\eta$, synchronization frequency $H$.
**Output:** Final trained model $\theta_T$.
1: **for** $t = 1, ..., T$ **do**
2:      **for** worker $m \in \mathcal{M}$ in parallel **do**
3:          $g_t^m(\theta_t^m) = 1/B \sum_{i=1}^{B} \nabla f_{x_{t,i} \sim \mathcal{D}}(\theta_t; x_{t,i})$;
4:          $\tilde{g}_t^m = 1/M \sum_{m \in \mathcal{M}} g_t^m(\theta_t^m) + \epsilon_t^m$; ▷ Communication
5:          $\theta_{t+1/2}^m = \theta_t^m - \eta_t \tilde{g}_t^m$;          ▷ Update model
6:          **if** $t + 1\%H = 0$ **then**
7:             $\theta_{t+1}^m = 1/M \sum_{m \in \mathcal{M}} \theta_{t+1/2}^m$;   ▷ Synchronization
8:          **else**
9:             $\theta_{t+1}^m = \theta_{t+1/2}^m$;
10: Return $\theta_T^m = \theta_T$;

---

### 4.1 PARAMETER SYNCHRONIZATION

To eliminate the model divergence $\Delta_t^m$, one intuitive approach is to directly synchronize model parameters across workers like (Lin et al., 2018). Specifically, after updating the model at iteration $t$, workers can communicate and average their parameters $\theta_{t+1}^m$, then reload the local models as $\bar{\theta}_{t+1}$. This synchronization ensures that the model divergence $\Delta_t^m$ is eliminated, setting it to zero. However, given the model size $S_\theta$, this synchronization per iteration incurs additional communication costs amounting to $TS_\theta$, which equals the original communication costs of the gradients. Therefore, reducing the overhead of parameter synchronization is crucial.

To address this, we propose `PAFT-Sync`, as detailed in Algorithm 1 and Fig. 3 (c). In addition to standard forward and backward propagation (FP and BP), gradient averaging, and model updating, `PAFT-Sync` averages model parameters after every $H$ training iteration. The model parameters are updated as follows:

$$\theta_{t+1}^m = \begin{cases} \theta_t^m - \eta_t \tilde{g}_t^m, & \text{if } t + 1\%H \neq 0 \\ \frac{1}{M} \sum_{m \in \mathcal{M}} (\theta_t^m - \eta_t \tilde{g}_t^m), & \text{if } t + 1\%H = 0 \end{cases}, \quad (7)$$

where $\tilde{g}_t^m = \bar{g}_t + \epsilon_t^m = \frac{1}{M} \sum_{m \in \mathcal{M}} g_t^m(\theta_t^m) + \epsilon_t^m$. After $H$ iterations, workers start training from the same point in the parameter space. The accumulated model divergence $\delta_t^m$ is cleared and re-accumulated at a low level, resulting in less harmful influences on gradient estimation. We theoretically and empirically demonstrate that this synchronization effectively eliminates the accumulated model divergence, thus ensuring training convergence.

**Definition 4.1.** (gap). The gap of a set $\mathcal{A} := \{a_0, a_1, ..., a_t\}$ of $t + 1$ integers, $a_i \leq a_{i+1}$ for $i = 0, ..., t - 1$, is defined as $\text{gap}(\mathcal{A}) := \max_{i=1,...,t}(a_i - a_{i-1})$.

Definition 4.1 is used to generally describe the fixed and dynamic synchronization frequency in both Algorithm 1 and 2. The timestamp in sequence $\{H_t\}$ represents the synchronization point. And the $\text{gap}(\{H_t\})$ is the maximal time gap between two synchronization points.

**Lemma 4.1.** *If $gap(\mathcal{A}) \leq H$ and sequence of decreasing positive stepsizes $\{\eta_t\}_{t \geq 0}$ satisfying $\eta_t \leq 2\eta_{t+H}$ for all $t \geq 0$, then. With the same initial point $\theta_0^m = \theta_0$ across workers $\{m | m = 1, 2, ..., M\}$, DSGD with noise $\epsilon_t^m \sim \mathcal{N}(0, \sigma^2)$ introduces accumulated model divergence $\Delta_t^m$ along the training process as*

$$\mathbb{E}||\bar{\theta}_{t+1} - \theta_{t+1}^m||^2 \leq \frac{4H(M+1)\sigma^2 \eta_t^2}{M} \quad (8)$$

**Remark.** Lemma 4.1 shows that the model divergence is bounded with $\mathcal{O}(H\sigma^2 \eta_t^2)$. Less $H$ helps to reduce this divergence, but introduces more communication overhead. In Section 4.2 shows that `PAFT-Dyn` finds a good trade-off between the convergence and the communication in Algorithm 2.

**Theorem 4.2.** *(Convergence with noised training with* `PAFT-Sync`*.) With object function defined in Eq. 1 satisfying Assumption 3.1, DSGD with* `PAFT` *(Eq. 7 or 10) noise* $\epsilon_t^m \sim \mathcal{N}(0, \sigma^2)$*, we have,*

$$
\mathbb{E}f(\hat{\theta}_T) - f^* \leq \frac{\mu a^3}{2 S_T}||\theta_0 - \theta^*||^2 + \frac{4T(T+2a)(\sigma_g^2 + \sigma^2)}{\mu M S_T}
$$
$$
+ \frac{256T}{\mu^2 S_T}\frac{(M+1)}{M}\sigma^2 HL
$$

(9)

*where* $\hat{\theta}_T = \frac{1}{MS_T}\sum_{m=1}^{M}\sum_{t=0}^{T-1} w_t \theta_t^m$*, for* $w_t = (a+t)^2$ *and* $S_T = \sum_{t=0}^{T-1} w_t \geq \frac{1}{3}T^3$

**Remark.** Theorem 4.2 shows that `PAFT` ensures the convergence of DSGD with noised gradients. And we can adjust the $H$ with respect to the noise variance $\sigma$ to trade off the convergence and communication. And Theorem 4.2 is dependent on a heterogeneous synchronization sequence $\{\mathcal{H}_t\}$ instead of a uniform sequence with the same gap $H$. Thus, it is general and can be easily extended to different algorithms that considering adjusting synchronization frequency.

**Corollary 4.3.** *Let* $\hat{\theta}_T$ *be defined as in Theorem 4.2, for parameter* $a = \max\{16\kappa, H\}$*. Then*

$$
\mathbb{E}f(\hat{\theta}_T) - f^* = \mathcal{O}\Big(\frac{\kappa^3 + H^3}{\mu T^3}\Big)G^2 + \mathcal{O}\Big(\frac{1}{\mu MT} + \frac{\kappa + H}{\mu MT^2}\Big)\sigma_g^2
$$
$$
+ \mathcal{O}\Big(\frac{(M+1)H\kappa}{\mu MT^2} + \frac{1}{\mu MT} + \frac{\kappa + H}{\mu MT^2}\Big)\sigma^2
$$

**Remark.** Corollary 4.3 shows that the convergence rate is the same as the SGD (Bottou et al., 2016).

## 4.2 ADJUSTING SYNCHRONIZATION FREQUENCY

While the synchronization can completely address the model divergence problem, it introduces extra communication overheads due to the communication of model parameters. Through the theoretical analysis (Theorem 4.2) in Section 4.1, we adjust the synchronization frequency $H$ detected error degrees of $\epsilon$ to reduce the unnecessary communication costs.

In light of this, we propose `PAFT-Dyn` in `PAFT`, as detailed in Algorithm 2. Compared with `PAFT-Sync` (Algorithm 1), `PAFT-Dyn` detects the magnitude of error degrees in training (Line 10) and adjusts $H_t$ according to $\sigma_t$ and the gradient norm (Line 11) to dynamically reduce communication costs.

Then, the new parameter synchronization scheme is given as follows.

$$
\theta_{t+1}^m = \begin{cases} \theta_t^m - \eta_t \tilde{g}_t^m, & \text{if } t+1 \notin \mathcal{H}_T \\ \frac{1}{M}\sum_{m \in \mathcal{M}}(\theta_t^m - \eta_t \tilde{g}_t^m), & \text{if } t+1 \in \mathcal{H}_T \end{cases}
$$

(10)

in which $\mathcal{H}_T$ is the sequence that indicates when to synchronize parameters.

---

**Algorithm 2** Distributed training with `PAFT`

**Input:** Initial model $\theta_0$, dataset $\mathcal{D}$, workers $\mathcal{M}$, total iteration $T$, learning rate $\eta$, initial detecting time gap $H_{\text{old}}$, initial synchronization sequence $\mathcal{H}_T = \{H_{\text{old}}\}$.
**Output:** Final trained model $\theta_T$.

1: **for** $t = 1, ..., T$ **do**
2:     **for** worker $m \in \mathcal{M}$ in parallel **do**
3:         $g_t^m(\theta_t^m) = \frac{1}{B}\sum_{i=1}^{B}\nabla f_{x_{t,i}\sim\mathcal{D}}(\theta_t; x_{t,i})$;
4:         $\tilde{g}_t^m = 1/M\sum_{m \in \mathcal{M}} g_t^m(\theta_t^m) + \epsilon_t^m$;
5:         **if** $t \in \mathcal{H}_T$ **then**
6:             $\theta_{t+1}^m = \theta_t^m - \eta_t \tilde{g}_t^m$;
7:             $\bar{\theta}_{t+1} = \frac{1}{M}\sum_{m \in \mathcal{M}}\theta_{t+1}^m$; (Async.)
8:         **else if** $t-1 \in \mathcal{H}_T$ **then**
9:             Wait for $\bar{\theta}_t = 1/M\sum_{m \in \mathcal{M}}\theta_t^m$;
10:           $\sigma_{\text{est}} = ||\bar{\theta}_{p,s} - \theta_{p,s}^m||$;
11:           $H_{\text{new}} = \text{All-Reduce}(||g_t^m||/\sigma_{\text{est}})$ ;
12:           Append $t + H_{\text{new}}$ in $\mathcal{H}_T$;
13:           $\theta_{t+1}^m = \bar{\theta}_t - \eta_t \tilde{g}_t^m$;
14:         **else**
15:           $\theta_{t+1}^m = \theta_t^m - \eta_t \tilde{g}_t^m$;
16: **Return** $\{\theta_T^m | m \in \mathcal{M}\}$;

---

**Estimating Error Degree.** The naive error detection method is directly computing the average of the gradients $1/M\sum_{m \in \mathcal{M}} g_t^m(\theta_t^m)$ and compare it with $\tilde{g}_t^m$ to estimate the noise degree of $\epsilon_t^m$, which introduces extra communication costs equal to synchronization. To this end, we estimate the error degree through the accumulated model divergence $\Delta_t^m$ to reduce the communication costs, as the $\Delta_t^m$ takes historical error information and need not be communicated at each iteration. According to Eq. 6 in Lemma 3.1, we can directly compute the accumulated model divergence $\Delta_t^m$ (Line 22 in Algorithm 2).

**Adjusting Synchronization Frequency.** Observing the convergence rate in Theorem 4.2, the intuitive way to adjust $H$ is set $H = \lceil 1/\sigma^2 \rceil$, thus the third term in the convergence bound (Eq. 9) becomes

as $\mathcal{O}(T(M+1)L/(MS_T))$. However, this too less $H$ actually is set too small and, because the dominant bound becomes as the second term as $\mathcal{O}(2T(T+2a)(\sigma_g^2 + \sigma^2)/(MS_T))$ and cannot be reduced by smaller $H$. Thus, we can set the $H = \sigma_g/\sigma$. Now, the second term and the third term in Eq. 9 is balanced. Note that the $H = \|g_{t,p_{max}}^m\|/\sigma_{max}$ also represents the signal-to-noise ratio (SNR) that is widely used in many methods to adjust hyper-parameters (Qiao et al., 2021).

### 4.3 OVERLAPPING SYNCHRONIZATION WITH TRAINING

Furthermore, synchronization after some training iterations still requires communication. To further reduce this communication cost, we overlap synchronization with the normal backward propagation process using asynchronous communication. The timeline of this overlapped communication is shown in Fig. 4.

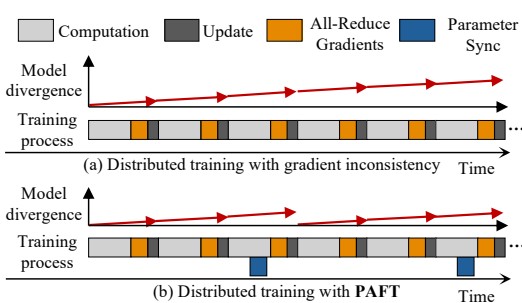

Figure 4: Overlapped synchronization with training.

As detailed in Algorithm 2, if the current round requires synchronization, the model averaging process is initiated without waiting (Line 7). In the next round, the model averaging can be overlapped with the forward and backward propagation processes. During model updating, workers wait for the previous round's synchronization to be completed. The new model parameters are then updated using the averaged model and the new gradients. Note that this approach introduces a trade-off, where we trade precise gradient estimation for the benefit of overlapping communication. We show the empirical effect on eliminating the model divergence in Appendix F.

### 4.4 EXTENSION TO OTHER OPTIMIZERS

The analysis in Seciton 3 is mainly built on the SGD, while the most of current DL models and LLMs are optimized with SGD momentum and Adam (Kingma & Ba, 2015). However, in the noised distributed training, the intrinsic characteristics of these optimizers are similar to the SGD. Specifically, the inconsistent gradients $gtil_t^m$ also lead to diverge updating directions of the model parameters, and the accumulated model divergence. Differently, the SGD momentum and Adam introduce extra terms including the momentum and precondition, which are updated according to the gradients. Thus, there is divergence existing in these extra terms. However, the divergence on them may not be accumulated as the model parameters as they are updated with moving averaging. Nevertheless, we can consider to synchronize these extra terms with the model parameters to ensure the convergence of the model. To this end, we provides results of synchronizing the momentum and precondition in Appendix F.

## 5 EXPERIMENTAL STUDIES

In this section, we conduct experiments on distributed training with varying degrees of noise. We compare basic distributed training without gradient inconsistency (Oracle), distributed training with gradient inconsistency (Noised), `PAFT-Sync` with different $H$ values, and `PAFT`.

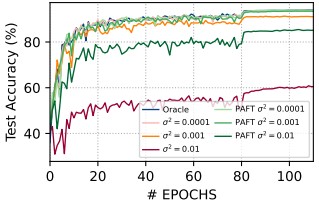 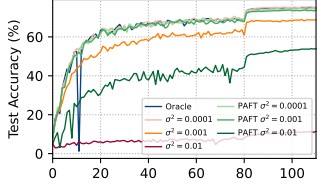 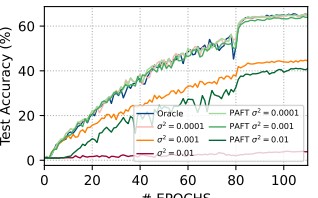

(a) Training ResNet-18 with 4 workers. (b) Training ResNet-50 with 4 workers. (c) Training ResNet-50 with 32 workers.

Figure 5: Different noise degrees.

**Cluster Configuration.** We have two testbeds including an 8-node GPU cluster, each of which installs 4 Nvidia RTX2080Ti GPU connected with PCIe3.0x16 with 10Gbps bandwidth, and a single GPU machine equipped with 8 Nvidia A6000 GPUs.

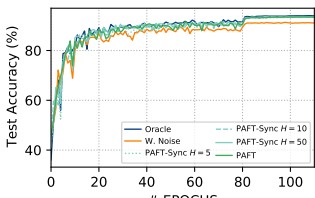 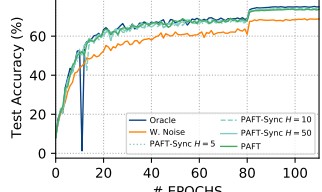 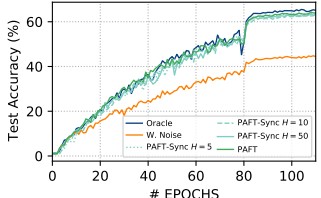

(a) Training ResNet-18 with 4 workers.  (b) Training ResNet-50 with 4 workers.  (c) Training ResNet-50 with 32 workers.

Figure 6: Different synchronization frequencies.

**DL Models and Datasets.** We train ResNet-18 (He et al., 2016) with CIFAR-10 (Krizhevsky et al., 2010), ResNet-50 (He et al., 2016) with CIFAR-100 with 120 epochs, and GPT-2 (Radford et al., 2019) with OpenWebText (Gokaslan et al., 2019) with 3K iterations. We also finetune pretrained LLaMA-2 (Touvron et al., 2023) and GPT-2 on Alpaca (Taori et al., 2023) using LoRA (Hu et al., 2021) with 1 epoch. ResNet-18 and ResNet-50 are optimized with SGDm (Bottou et al., 2016) with learning rate of 0.1 and momentum of 0.9. GPT-2 is trained with Adam (Kingma & Ba, 2015) with learning rate of 0.001, $\beta_1$ as 0.9 and $\beta_2$ as 0.99.

**Simulation of Gradient Inconsistency.** We simulate the noise with different degrees by adjusting $\sigma$ with range $\{0.0001, 0.001, 0.01, 0.1\}$. The small noise degree $\{0.0001, 0.001\}$ can represent the small communication noises. While the larger noise $\{0.01, 0.1\}$ can simulate the bit corruptions or the large communication noise, which appears less during training.

### 5.1 MAIN RESULTS

Fig. 5(a) and 5(b) show convergence of noised distributed training on ResNet-18 and ResNet-50 with 4 workers. Fig. 5(c) show training resnet-50 of noised distributed training with 32 workers. All results show that as noise degree increases, the accuracy of model declines correspondingly. While PAFT can successfully illuminate the small noise influence and mitigate the large noise influence.

The results in all figures show that the PAFT can successfully defend against noise and improve the convergence of noised training when $\sigma^2 = 0.0001$ or $0.001$. Note that there is still gap between the normal training (Oracle) and PAFT when $\sigma^2 \geq 0.01$. The reason is that the noise not only introduces gradient inconsistency, but also the noised gradient direction that influences gradient descend. This is the inherent problem of the noise, like the Byzantine Fault-tolerance problem (Guerraoui et al., 2024).

**Training and Finetuning LLMs.** Fig. 7, 8(a) and 8(b) show the loss curves of pretraining and fine-tuning

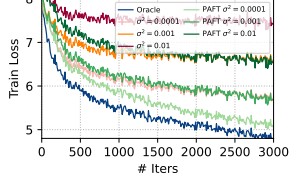 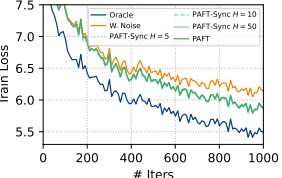

(a) Different noise degrees.  (b) Different Sync. frequency.

Figure 7: Training GPT-2 with OpenWebText.

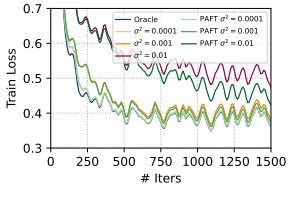 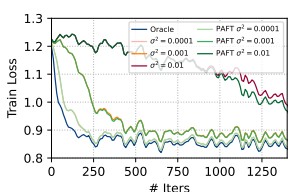

(a) GPT-2 with Alpaca.  (b) LLaMA-2 with Alpaca.

Figure 8: Finetuning LLMs with different noise degrees.

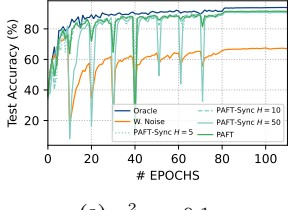 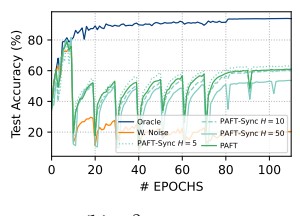

(a) $\sigma^2_{\text{large}} = 0.1$.  (b) $\sigma^2_{\text{large}} = 1.0$.

Figure 9: Training ResNet-18 with accidental large noise.

LLMs. The results show that the PAFT can successfully defend against noise and improve the convergence. While the model size increases from ResNets to LLMs like GPT-2 and LLaMA-2, the PAFT can significantly improve than baselines. When the noise degree $\sigma^2 = 0.0001$ or $0.001$, the PAFT can almost ensure the convergence as similar to the training without noise. While for the larger noise $\sigma^2 = 0.01$, the PAFT can improve the convergence compared with the noised training. The exiting

performance gap between `PAFT` and the normal training without noise comes from the noisy gradient itself, which leads to an incorrect updating direction. Future works should consider combining both synchronization and voting mechanisms like the Byzantine Fault-tolerance problem (Guerraoui et al., 2024) to address this problem.

**Accidental Large Noise.** We simulate accidental large noise, like bit corruptions. Specifically, in each round, the noise is sampled from $\mathcal{N}(0, 0.0001)$ to simulate normal small noises. After each 500 iterations, the noise is sampled from a $\mathcal{N}(0, 0.1)$ or $\mathcal{N}(0, 1.0)$ as simulated accidental large noise.

The Fig. 9(a) shows training with large noise sampled from $\mathcal{N}(0, 0.1)$ while Fig. 9(b) shows $\mathcal{N}(0, 1.0)$. The convergence curves clearly demonstrate the influence of this accidental noise. In each iteration that the noise happens, the test accuracy instantly drops a lot and is pulled back by `PAFT` from the valley. However, for a large noise with variance of $1.0$, it is hard to pull it back. Interestingly, we observe that the learning rate decay at the late stage helps the model defend against the noise. Less learning rate results in less model update and divergence, which aligns with our theoretical analysis (Lemma 3.1 and Theorem 3.2).

**Wall-clock Iteration Time** We provide a comparison of the average iteration wall-clock time (in seconds) during the training of the ResNet-50 model, using different numbers of workers ranging from $4 \sim 32$ in Table 1. By dynamic adjusted synchronization frequency and overlapped communication, the `PAFT` reduces the extra cost

Table 1: Average iteration wall-clock time (seconds) of training ResNet-50.

| # of workers | 4 | 8 | 16 | 32 |
|---|---|---|---|---|
| DSGD | 0.201 | 0.212 | 0.228 | 0.333 |
| PAFT-Sync | 0.243 | 0.254 | 0.276 | 0.411 |
| PAFT | 0.237 | 0.244 | 0.253 | 0.373 |

than `PAFT-Sync` for around up to 11.0% efficiency improvement for 32 workers. And the extra cost of `PAFT` than DSGD is around 18.9% for 32 workers. For more workers, `PAFT-Sync` shows better improvement, which means the good scalability of `PAFT-Sync`.

## 6 RELATED WORKS

We provide a concise literature review here due to the limited space. We provide detailed related works in Appendix C.

**Parallelism at Scale** Distributed LM training leverages hybrid parallelism (DP, TP, PP) (Narayanan et al., 2021). DP scales training via batch size increases and model replication (Krizhevsky et al., 2017; Tang et al., 2020), while TP (Narayanan et al., 2021) and PP (Narayanan et al., 2019; Rasley et al., 2020) address single-device memory constraints. `PAFT` resolves gradient aggregation (GA) errors and integrates with frameworks like DeepSpeed (Rasley et al., 2020) and Megatron (Narayanan et al., 2021) for scalable LLM training.

**Safety and Reliability** Existing work ensures reliability via checkpointing (Wang et al., 2023b; Narayanan et al., 2021), elasticity (Thorpe et al., 2022), and Byzantine fault tolerance (El-Mhamdi et al., 2020; Guerraoui et al., 2024). GA errors, caused by hardware/communication faults, are distinct for their subtlety and detection complexity. `PAFT` is the first to mitigate GA errors at scale.

## 7 CONCLUSION

In this work, we address GA errors in distributed training caused by hardware issues like bit corruptions and communication noise, which are challenging to capture and mitigate for fault tolerance. We first mathematically formulate and generalize these errors as gradient inconsistency. Then, we theoretically analyze how they lead to accumulated model divergence and failed convergence. To address this issue, we propose `PAFT`, a fault-tolerant distributed training system incorporating dynamic and asynchronous parameter synchronization optimizations. The two components of `PAFT-Sync` and `PAFT-Dyn` work synergistically to mitigate the harm of GA errors. `PAFT-Sync` maintains model convergence by periodically synchronizing parameters, while `PAFT-Dyn` minimizes overhead by adjusting synchronization frequency based on the profiled error degrees. Our implementation of `PAFT` on PyTorch Distributed, evaluated on ResNet-18, ResNet-50, GPT-2, and LLaMA-2 models across 32 GPUs, demonstrates the system's robustness against a wide range of GA errors. Evaluation results indicate that, unlike vanilla distributed training, `PAFT` effectively maintains fault tolerance without compromising training throughput. Thus, `PAFT` offers a scalable and efficient solution for improving reliability of large-scale distributed training systems in the presence of GA errors.

ETHICS STATEMENT

We declare no conflicts of interest that could inappropriately influence our work. Our study does not involve human subjects, data collection from individuals, or experiments on protected groups. The models and datasets used are publicly available and widely used in the research community. We have made efforts to ensure our experimental design and reporting of results are fair, unbiased, and do not misrepresent the capabilities or limitations of the methods presented.

REPRODUCIBILITY STATEMENT

The experiments are easy to reproduce with Pytorch Distributed framework. And we declare that we will open-source our code for reproducibility. Also, we have provided all details of hyperparameters and experiment settings in Section 5.

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

APPENDIX

## A  THE USE OF LARGE LANGUAGE MODELS

We used LLMs solely for grammar and wording improvements. It did not generate ideas, analyses, or results. No additional or undisclosed LLM use occurred.

## B  LIMITATIONS

**Replacement of model synchronization.** There are some other ways to keep the models synchronized without extra effort. For example, the FSDP (Paszke et al., 2017) or deepspeed zero2/3 (Artetxe et al., 2021) synchronize the model parameters during training. Thus, it seems that PAFT can be replaced by using these techniques.

While FSDP or DeepSpeed Zero 2/3 can indeed synchronize model parameters, they lack redundancy of new model parameters for possible elastic fault tolerance from model replication in DDP. In scenarios where worker failures occur, the FSDP must reload previous model parameters and repeat current iterations, which disrupts elastic training. This means that training cannot simply resume from other duplicated parameters like in  (Wan et al., 2025) in the event of a single-point failure. Nevertheless, the main contribution of this work are identifying and analyzing how the gradient aggregation error leads to failed convergence, and model synchronization can help mitigate this kind of error.

**Errors happening during model synchronization.** If SDC could happen in gradient synchronization, then it could also happen in model synchronization/averaging. In this work, we assume that model synchronization/averaging is always safe and correct without SDC. Considering this problem, to strictly ensure guaranteed parameter synchronization, potential solutions are fault-tolerant communication protocols for model synchronization. However, this may incur higher communication costs and lower efficiency compared to high-speed communication protocols in data centers. Since model synchronization occurs less frequently in PAFT, it takes less costs than guaranteed gradient aggregation.

## C  MORE RELATED WORKS

### C.1  PARALLELISM AT SCALE

Distributed large model (LM) training (Narayanan et al., 2021) employs hybrid parallelism techniques, including data parallelism, tensor model parallelism, and pipeline parallelism.

**Data parallelism (DP)** (Krizhevsky et al., 2017; Chen et al., 2016; Cui et al., 2016; Zhang et al., 2017; Tang et al., 2020; 2022), which replicates models for parallel training, is central in hybrid parallelism. It scales the training effectively by increasing the batch size to accelerate model convergence. However, DP is limited by memory capacity and communication overheads, especially for large-scale LM training. This paper focuses on the GA erros in DP training.

**Tensor model parallelism (TP)** (Or et al., 2020; Narayanan et al., 2021) complement DP by addressing memory limitations when models exceed a single device's memory capacity. `PAFT` tackles GA errors and has been generalized to hybrid parallel training frameworks like DeepSpeed (Rasley et al., 2020) and Megatron (Narayanan et al., 2021) towards large-scale LM training. The TP training may also have communication errors, which are out of the scope of this paper. And the communication errors in concatenating tensors in TP are more like the computational SDC errors, which is different from the GA errors in DP.

**Pipeline parallelism (PP)** (Narayanan et al., 2019; Rasley et al., 2020; Tang et al., 2023) splits the whole model into different stages and processes them in a pipelined manner. The PP can reduce the memory consumption and communication overheads. The communication errors in PP are more like the quantization or compression errors, which are different from the GA errors, either.

## C.2 SAFETY AND RELIABILITY OF DISTRIBUTED TRAINING

**Active Failures.** Many studies focus on system reliability concerning node failures, which may directly interrupt training processes. These studies propose fault-tolerant mechanisms using check-pointing (Wang et al., 2023b; 2024; Narayanan et al., 2021) and elasticity (Thorpe et al., 2022; Harlap et al.; He et al., 2023a) optimizations for rapid recovery. These optimizations enhance system robustness and enable quick restarts.

**Silent Failures.** There are other soft failures like the communication noise happen in GA, or the workers upload the wrong gradients to the server. The typical methods to handle these failures include gradient clip, or considering them as the Byzantine faults by malicious node behavior (El-Mhamdi et al., 2020; Damaskinos et al., 2018; Guerraoui et al., 2024). However, the silent errors in GA errors in the scope of this paper, arise from unintentional issues like hardware errors or communication errors, leading to inaccuracies in gradient updates. And we mainly focus on the GA errors happen during broadcasting in DP training, which is different from the other types of soft failures.

## C.3 ASYNCHRONOUS OPTIMIZATIONS

To accelerate distributed training, asynchronous optimization techniques have been proposed to reduce the synchronization overheads (Tsitsiklis et al., 1986; Zheng et al., 2017; Damaskinos et al., 2018). These techniques allow workers to update model parameters independently, reducing the waiting time for synchronization. To consider accelerating synchronizing checkpoints, many works utilize the asynchronous and heterogeneous capabilities of hardware resources for parallel processing of different tasks. For example, in checkpointing optimizations, asynchronous parameter snapshotting can compete for memory bandwidth with training processes, potentially slowing down the training speed (Mohan et al.; team, 2022; Wang et al., 2024). Additionally, inter-node communications asynchronous to training can introduce communication overheads (Shi et al., 2020; 2021b). In PAFT-Sync, we also observe unavoidable asynchronous overheads during training. However, the dynamic synchronization frequency effectively reduces the overall asynchronous overhead in the fault-tolerant system.

## C.4 FAULT TOLERANCE IN FEDERATED LEARNING

Some works investigate fault tolerance of federated learning (FL). Considering the impact of unreliable devices (e.g., dropouts, misconfigurations, poor data quality) on FL performance, especially in rural environments with limited clients, how infrastructure-level errors (e.g., unstable power, network issues) and ML-specific inconsistencies (e.g., misconfigured hyperparameters, low-quality data) affect FL model accuracy is not explored (Huang et al., 2023). It is found that FedAVG can perform well even with unreliable clients (Huang et al., 2023). This shows that the parameter synchronization might be a useful tool in different kinds of fault-tolerance scenarios.

Some works (Mansouri et al., 2022) propose a new secure and fault-tolerant aggregation scheme that can recover from client failures. Their solution is based on a threshold variant of the Joye-Libert secure aggregation scheme, combined with decentralized key management and input encoding. The scheme allows a set of available clients to compute the encryption of a zero value on behalf of missing clients, enabling the aggregator to correctly aggregate the inputs of the online clients.

And some work (Fan et al., 2021) proposes a novel FRL framework, Federated Policy Gradient with Byzantine Resilience (FedPG-BR), which ensures convergence and tolerates a certain percentage of faulty agents. The theoretical analysis of FedPG-BR demonstrates its improved sample efficiency with more agents and its resilience to Byzantine faults. The key idea is to design a gradient-based Byzantine filter on top of a variance-reduced federated policy gradient framework. Some works improve the homomorphic encryption considering the fault-tolerance (Zhang et al., 2024).

Byzantine fault-tolerant FL is a mainly focused direction (Liu et al., 2021). The redundancies in FL agents' cost functions are necessary and sufficient to ensure Byzantine resilience. Using a root dataset by the service provider (Cao et al., 2021) helps to detect the malicious attacks.

| Component | Category | Interruption Count | % of Interruptions |
|-----------|----------|--------------------|--------------------|
| Faulty GPU | GPU | 148 | 30.1% |
| GPU HBM3 Memory | GPU | 72 | 17.2% |
| Software Bug | Dependency | 54 | 12.9% |
| Network Switch/Cable | Network | 35 | 8.4% |
| Host Maintenance | Unplanned Maintenance | 32 | 7.6% |
| GPU SRAM Memory | GPU | 19 | 4.5% |
| GPU System Processor | GPU | 17 | 4.1% |
| NIC | Host | 7 | 1.7% |
| NCCL Watchdog Timeouts | Unknown | 7 | 1.7% |
| Silent Data Corruption | GPU | 6 | 1.4% |
| GPU Thermal Interface + Sensor | GPU | 6 | 1.4% |
| SSD | Host | 3 | 0.7% |
| Power Supply | Host | 3 | 0.7% |
| Server Chassis | Host | 2 | 0.5% |
| IO Expansion Board | Host | 2 | 0.5% |
| Dependency | Dependency | 2 | 0.5% |
| CPU | Host | 2 | 0.5% |
| System Memory | Host | 2 | 0.5% |

Table 2: **Root-cause categorization of unexpected interruptions during a 54-day period of Llama-3 405B pre-training.** (Dubey et al., 2024) About 78% of unexpected interruptions were attributed to confirmed or suspected hardware issues.

Table 3: Type of GPU Xid Errors and Its Causes

| Xid Errors | Analysis |
|-----------|----------|
| Software Causes: Xid_13/31 Xid_43/45 | Triggered by application programs, software-related Xid messages may indicate anomalies in GPU memory affecting code and data segments. However, it's crucial to consider other information for a comprehensive hardware functionality assessment. |
| NVLink Error: Xid 74 | Xid74 indicates errors in NVLink. For PCIe A100, it mainly occurs on the NVLink Bridge between two GPUs. Its occurrence rate is several orders of magnitude higher than other hardware faults. Apart from stress testing to exclude those that are constantly repeating errors, there isn't a good way to avoid the occurrence of Xid74 issues. |
| Memory ECC Error: Xid_63/64 Xid_94/95 | Triggered when the GPU handles memory ECC errors on the GPU. With the introduction of row remapping technology in A100, most instances can be resolved by simply resetting the GPU to retain optimal performance. |
| Uncorrectable GPU Failures: Xid_44/48 Xid_61/62/69/79 | These failures mean an uncorrectable error occurs on the GPU, which is also reported back to the user application. A GPU reset or node reboot is needed to clear this error. |
| Other Failures: Xid 119 | Xid119 means GPU GSP module failed. These failures need to undergo a field test, and most need to RMA. |

## D  MORE DISCUSSION

### D.1  SILENT DATA CORRUPTION ERRORS

Silent data corruption (SDC) errors are particularly insidious in high-performance computing (HPC) (Wang et al., 2023a; He et al., 2023b), database (Bacon, 2022) and communication systems because they can go undetected and lead to incorrect results. These errors can occur due to various reasons, including hardware faults, software bugs, or cosmic radiation. In the context of HPC, SDC errors can significantly impact the reliability and accuracy of computations, especially

Table 4: **Raw Data** of GPU Xid Errors during one year in Fire-Flyer HPC (An et al., 2024)

| GPU Error Type | Xid Code | Number | Percentage |
|---|---|---|---|
| NVLink Error | xid_74 | 5521 | 42.57% |
| Software Causes | xid_13 | 45 | 0.35% |
| | xid_31 | 2487 | 19.18% |
| | xid_43 | 4342 | 33.48% |
| | xid_45 | 240 | 1.85% |
| GPU ECC Error | xid_63 | 245 | 1.89% |
| | xid_64 | 2 | 0.02% |
| | xid_94 | 13 | 0.10% |
| | xid_95 | 17 | 0.13% |
| Uncorrectable Failures | xid_44 | 1 | 0.01% |
| | xid_48 | 2 | 0.02% |
| | xid_61 | 13 | 0.10% |
| | xid_62 | 3 | 0.02% |
| | xid_69 | 1 | 0.01% |
| | xid_79 | 37 | 0.29% |
| GPU GSP ERROR | xid_119 | 1 | 0.01% |
| **Total** | | **12970** | **100.00%** |

in large-scale simulations and data-intensive applications. The large-scale distributed deep learning might be severely influenced by the SDC errors (He et al., 2023b).

In communication systems, SDC errors can be introduced during data transmission between nodes in a distributed computing environment (Fiala et al., 2012; łgorzata Steinder & Sethi, 2004). These errors can be caused by issues such as faulty network hardware, electromagnetic interference, or signal degradation over long distances. The impact of SDC errors in communication can be severe, as they can lead to incorrect data being propagated through the system, potentially causing widespread computational errors.

Table 2 shows the root-cause categorization of unexpected interruptions during a 54-day period of Llama 3 405B pre-training (Dubey et al., 2024). About 78% of unexpected interruptions were attributed to confirmed or suspected hardware issues, including faulty GPUs, GPU memory, and other components. These interruptions can lead to significant downtime and data loss, affecting the overall performance and reliability of the system. The SDC and network errors occupy a significant portion of the interruptions, highlighting the importance of addressing these issues in distributed computing environments. Note that the reported SDC erros belong to the explicit results that are obviously observed. However, there exists a large portion of silent errors with low error degree may appear in the training process, which is hard to detect and not reported.

There is a substantial amount of SDC in data center processors (He et al., 2023b; Wang et al., 2023a), leading to complex issues that are difficult to replicate and locate. In Fire-Flyer HPC (An et al., 2024), various computational errors and GPU memory errors not detected by Error Correction Code (ECC) listed in Table 3, which led to models' gradnorm spikes, loss explosions, and even nonconvergence. Tackling these silent errors is crucial for ensuring the reliability and accuracy of distributed training systems. The errors like Xid 63/64 will cause the failed convergence problems.

Table 4 shows that NVLink Erros and software errors occupy a large portion of all errors. It is crucial to address the SDC erros in both communication and computation.

## D.2 SDC ERROR SIMULATION

Fig. 10 shows the bias distribution with different noise degrees. For the $\sigma = 0.0001$, almost all elements are less than 3e-4. Fig. 11 shows the maximal value in the noise during training with different noise degrees. After each 500 iterations, there is a burst value happens, which is more significant for the larger noise degree.

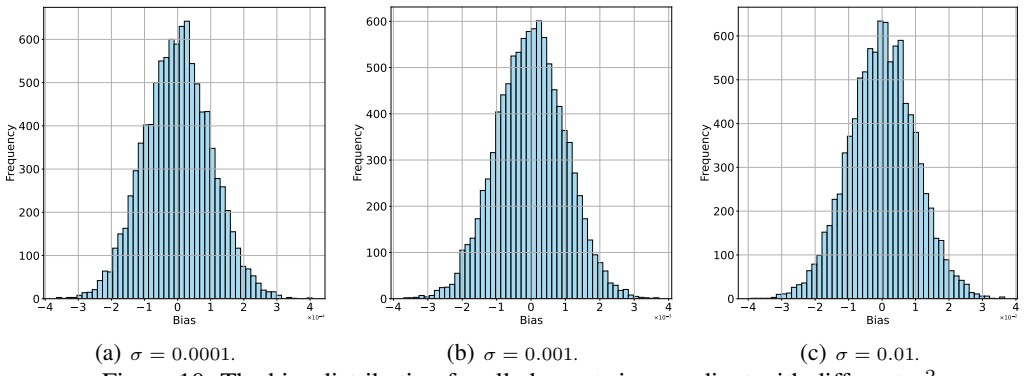

(a) $\sigma = 0.0001$.      (b) $\sigma = 0.001$.      (c) $\sigma = 0.01$.

Figure 10: The bias distribution for all elements in a gradient with different $\sigma^2$.

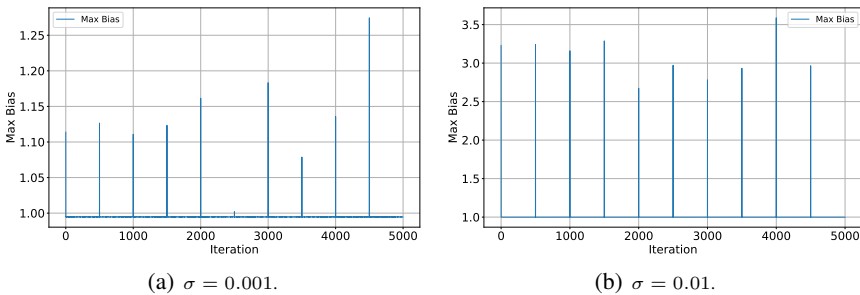

(a) $\sigma = 0.001$.      (b) $\sigma = 0.01$.

Figure 11: The maximal value in the noise during training.

**Gradient Magnitude Distribution.** Fig. 12 and 13 show the distribution of values in the gradients of the ResNet-50 when training with CIFAR-100 at different iterations. Comparing the magnitudes of gradients with the noise, we can see that even the noise with $\sigma = 0.001$ is a large noise that has similar magnitude to the gradients. In real-world scenarios, noises with $\sigma \geq 0.01$ happen less. For the significantly larger noise error, which can be detected by some machine learning methods like the gradient clip, or the majority voting.

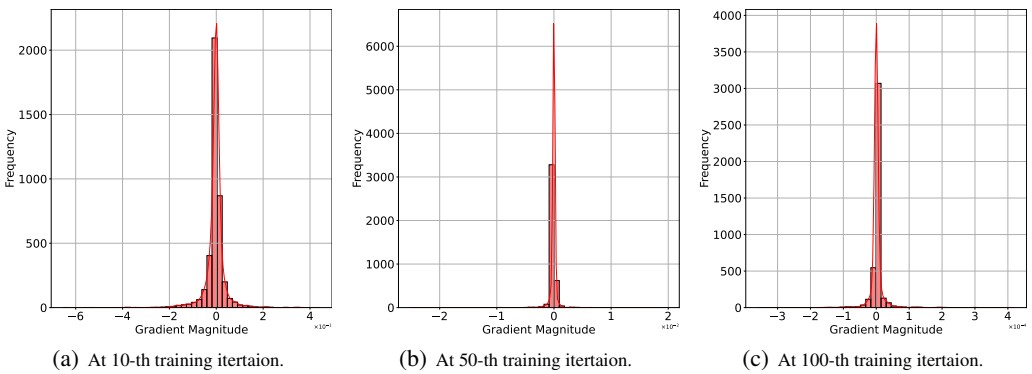

(a) At 10-th training itertaion.      (b) At 50-th training itertaion.      (c) At 100-th training itertaion.

Figure 12: The bias distribution for all elements of gradients of Conv layer in the first block.

## E PROOF

In this section, we provide the detailed proof of Lemma 3.1, Theorem 3.2, Lemma 4.1 and Theorem 4.2. We rewrite all of them in this section for the convenience of reading.

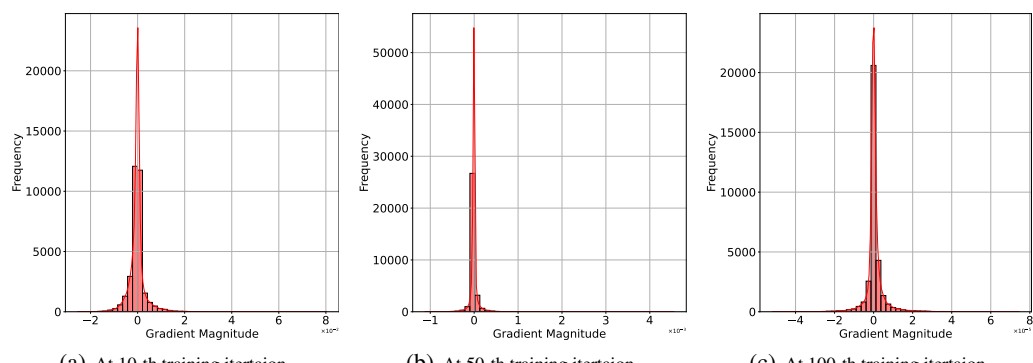

(a) At 10-th training itertaion.    (b) At 50-th training itertaion.    (c) At 100-th training itertaion.

Figure 13: The bias distribution for all elements of gradients of Conv layer in the second block.

### E.1 SOME DEFINITIONS AND ASSUMPTIONS

**Definition E.1.** (*Virtual Average*). In distributed stochastic gradient descend (Eq. 5) with inconsistent gradient (Definition 2.1), an averaged model weight sequence $\{\bar{\theta}_t\}_{t\geq0}$ is defined as

$$\bar{\theta}_0 = \theta_0, \qquad\qquad \bar{\theta}_t = \frac{1}{M}\sum_{i=1}^{M}\theta_t^i. \qquad (11)$$

From Definition 2.1, Eq. 5 and 11, we have

$$\bar{\theta}_{t+1} = \bar{\theta}_t - \eta_t\tilde{g}_t. \qquad (12)$$

### E.2 INCREASING MODEL DIVERGENCE

**Lemma E.1** (Increasing Model Divergence (Lemma 3.1)). *With the same initial point $\theta_0^m = \theta_0$ across workers $\{m|m = 1, 2, ..., M\}$, the DSGD with noise $\epsilon_t^m \sim \mathcal{N}(0, \sigma^2)$ introduces accumulated model divergence $\Delta_t^m$ along the training process as*

$$\mathbb{E}||\bar{\theta}_{t+1} - \theta_{t+1}^m||^2 = \frac{(M+1)\sigma^2}{M}\sum_{s=0}^{t}\eta_s^2. \qquad (13)$$

*Proof of Lemma 3.1.* We define the $\bar{\theta}_t = \frac{1}{M}\sum_{i=1}^{M}\theta_t^i$ and $\tilde{g}_t = \frac{1}{M}\sum_{i=1}^{M}\tilde{g}_t^i = \frac{1}{M}\sum_{i=1}^{M}(\bar{g}_t + \epsilon_t^m)$. Then, we have $\bar{\theta}_{t+1} = \bar{\theta}_t - \eta_t\tilde{g}_t$. By substituting Eq. 4 and 5 into $\Delta_t^m$ and iterating.

$$\begin{aligned}
\mathbb{E}||\bar{\theta}_{t+1} - \theta_{t+1}^m||^2 &= \mathbb{E}||\bar{\theta}_t - \eta_t\tilde{g}_t - \theta_t^m + \eta_t\tilde{g}_t^m||^2 \\
&= \mathbb{E}||\bar{\theta}_t - \theta_t^m||^2 + \eta_t^2\mathbb{E}||\tilde{g}_t - \tilde{g}_t^m||^2 \\
&\quad + 2\eta_t\underbrace{\mathbb{E}\langle\bar{\theta}_t - \theta_t^m, \tilde{g}_t^m - \tilde{g}_t\rangle}_{=0}.
\end{aligned} \qquad (14)$$

By iterating above equation from $t \to 0$, we have

$$\begin{aligned}
\mathbb{E}||\bar{\theta}_{t+1} - \theta_{t+1}^m||^2 &= \underbrace{\mathbb{E}||\bar{\theta}_0 - \theta_0^m||^2}_{=0} + \sum_{s=0}^{t}\eta_s^2\mathbb{E}||\tilde{g}_s - \tilde{g}_s^m||^2 \\
&= \sum_{s=0}^{t}\eta_s^2\text{Var}(\frac{1}{M}\sum_{k=1}^{M}(\bar{g}_t + \epsilon_t^k) - (\bar{g}_t + \epsilon_t^m)) \\
&= \sum_{s=0}^{t}\eta_s^2\text{Var}(\frac{1}{M}\sum_{k=1}^{M}\epsilon_s^k - \epsilon_s^m) \\
&= \frac{(M+1)\sigma^2}{M}\sum_{s=0}^{t}\eta_s^2
\end{aligned}$$

$\square$

### E.3 Convergence with noised training.

Firstly, we provide the Lemma E.2 before proving Theorem 3.2.

**Lemma E.2.** *Let $\{\theta_t\}_{t\geq 0}$ and $\{\bar{\theta}_t\}_{t\geq 0}$ for $m \in [M]$ be defined as in Equation (7), (11) and let f be L-smooth and $\mu$-strongly convex and $\eta_t \leq \frac{1}{4L}$. Then*

$$
\mathbb{E}||\bar{\theta}_{t+1} - \theta^*||^2 \leq (1 - \mu\eta_t)\mathbb{E}||\bar{\theta}_t - \theta^*||^2 + \eta_t^2\mathbb{E}||\tilde{g}_t - \nabla F_t||^2
$$

$$
- \frac{1}{2}\eta_t\mathbb{E}(f(\bar{\theta}_t) - f^*) + \frac{2L\eta_t}{M}\sum_{i=1}^{M}\mathbb{E}||\bar{\theta}_t - \theta_t^i||^2 \tag{15}
$$

*Proof of Lemma E.2.* Using the update Equation 12, we have

$$
\begin{aligned}
||\bar{\theta}_{t+1} - \theta^*||^2 &= ||\bar{\theta}_t - \eta_t\tilde{g}_t - \theta^*||^2 \\
&= ||\bar{\theta}_t - \eta_t\tilde{g}_t - \theta^* - \eta_t\nabla F_t + \eta_t\nabla F_t||^2 \\
&= ||\bar{\theta}_t - \eta_t\nabla F_t - \theta^*||^2 + \eta_t^2||\tilde{g}_t - \nabla F_t||^2 \\
&\quad + 2\eta_t\langle\bar{\theta}_t - \theta^* - \eta_t\nabla F_t, \tilde{g}_t - \nabla F_t\rangle.
\end{aligned} \tag{16}
$$

Observe that

$$
\begin{aligned}
&||\bar{\theta}_t - \eta_t\nabla F_t - \theta^*||^2 \\
&= ||\bar{\theta}_t - \theta^*||^2 + \eta_t^2||\nabla F_t||^2 - 2\langle\bar{\theta}_t - \theta^*, \eta_t\nabla F_t\rangle \\
&\leq ||\bar{\theta}_t - \theta^*||^2 + \eta_t^2\frac{1}{M}\sum_{i=1}^{M}||g(\theta_t^i)||^2 \\
&\quad - \frac{2\eta_t}{M}\sum_{i=1}^{M}\langle\bar{\theta}_t - \theta_t^i + \theta_t^i - \theta^*, g(\theta_t^i)\rangle \\
&= ||\bar{\theta}_t - \theta^*||^2 + \eta_t^2\frac{1}{M}\sum_{i=1}^{M}||g(\theta_t^i) - g(\theta^*)||^2 \\
&\quad - \frac{2\eta_t}{M}\sum_{i=1}^{M}\langle\theta_t^i - \theta^*, g(\theta_t^i)\rangle - \frac{2\eta_t}{M}\sum_{i=1}^{M}\langle\bar{\theta}_t - \theta_t^i, g(\theta_t^i)\rangle.
\end{aligned} \tag{17}
$$

By $L$-smoothness, we have

$$
||g(\theta_t^i) - g(\theta^*)||^2 \leq 2L(f(\theta_t^i) - f^*). \tag{18}
$$

By $\mu$-strong convexity, we have

$$
-\langle\theta_t^i - \theta^*, g(\theta_t^i)\rangle \leq -(f(\theta_t^i) - f^*) - \frac{\mu}{2}||\theta_t^i - \theta^*||^2. \tag{19}
$$

To estimate the last term in (17), we use $2\langle a, b\rangle \leq \gamma||a||^2 + \gamma^{-1}||b||^2$ for $\gamma > 0$, thus

$$
\begin{aligned}
-2\langle\bar{\theta}_t - \theta_t^i, g(\theta_t^i)\rangle &\leq 2L||\bar{\theta}_t - \theta_t^i||^2 + \frac{1}{2L}||g(\theta_t^i)||^2 \\
&= 2L||\bar{\theta}_t - \theta_t^i||^2 + \frac{1}{2L}||g(\theta_t^i) - g(\theta^*)||^2 \\
&\leq 2L||\bar{\theta}_t - \theta_t^i||^2 + (f(\theta_t^i) - f^*).
\end{aligned} \tag{20}
$$

By applying these estimates to (17), we get

$$
\begin{aligned}
&||\bar{\theta}_t - \theta^* - \eta_t\nabla F_t||^2 \\
&\leq ||\bar{\theta}_t - \theta^*||^2 + \frac{2\eta_t L}{M}\sum_{i=1}^{M}||\bar{\theta}_t - \theta_t^i||^2 \\
&\quad + \frac{2\eta_t}{M}\sum_{i=1}^{M}((\eta_t L - \frac{1}{2})(f(\theta_t^i) - f^*) - \frac{\mu}{2}||\theta_t^i - \theta^*||^2)
\end{aligned} \tag{21}
$$

For $\eta_t \leq \frac{1}{4L}$ it holds $(\eta_t L - \frac{1}{2}) \leq -\frac{1}{4}$. By convexity of $a(f(\theta) - f^*) + b||\theta - \theta^*||^2$ for $a, b \geq 0$,

$$- \frac{1}{M} \sum_{i=1}^{M} (a(f(\theta_t^i) - f^*) + b||\theta_t^i - \theta^*||^2) \tag{22}$$

$$\leq - (a(f(\bar{\theta}_t) - f^*) + b||\bar{\theta}_t - \theta^*||^2).$$

Hence, we can continue in (21) and obtain

$$||\bar{\theta}_t - \theta^* - \eta_t \nabla F_t||^2 \leq (1 - \mu\eta_t)||\bar{\theta}_t - \theta^*||^2$$

$$- \frac{1}{2}\eta_t(f(\bar{\theta}_t) - f^*) + \frac{2\eta_t L}{M} \sum_{i=1}^{M} ||\bar{\theta}_t - \theta_t^i||^2 \tag{23}$$

Finally, we can combine (23) with (16). By taking expectation, we get

$$\mathbb{E}||\bar{\theta}_{t+1} - \theta^*||^2 \leq (1 - \mu\eta_t)\mathbb{E}||\bar{\theta}_t - \theta^*||^2 + \eta_t^2 \mathbb{E}||\tilde{g}_t - \nabla F_t||^2$$

$$- \frac{1}{2}\eta_t \mathbb{E}(f(\bar{\theta}_t) - f^*) + \frac{2L\eta_t}{M} \sum_{i=1}^{M} \mathbb{E}||\bar{\theta}_t - \theta_t^i||^2 \tag{24}$$

$\square$

Now, we can prove Theorem 3.2 with the help of Lemma E.2.

**Theorem E.3** (Convergence with noised training (Theorem 3.2.)). *With object function defined in Eq. 1 satisfying Assumption 3.1, DSGD with noise $\epsilon_t^m \sim \mathcal{N}(0, \sigma^2)$ has the following convergence bound*

$$\frac{1}{T} \sum_{t=0}^{T-1} \eta_t \mathbb{E}(f(\bar{\theta}_t) - f^*) \leq \frac{2\mathbb{E}||\bar{\theta}_0 - \theta^*||^2}{T} + \frac{2(\sigma_g^2 + \sigma^2)}{TM} \sum_{t=0}^{T-1} \eta_t^2$$

$$+ \frac{4L\sigma^2(M + 1)}{TM} \sum_{t=0}^{T-1} \eta_t \sum_{s=0}^{t-1} \eta_s^2.$$

### E.4 BOUNDED MODEL DIVERGENCE

**Lemma E.4** (Bounded Model Divergence (Lemma 4.1)). *If $gap(\mathcal{A}) \leq H$ and sequence of decreasing positive stepsizes $\{\eta_t\}_{t \geq 0}$ satisfying $\eta_t \leq 2\eta_{t+H}$ for all $t \geq 0$, then. With the same initial point $\theta_0^m = \theta_0$ across workers $\{m|m = 1, 2, ..., M\}$, the DSGD with noise $\epsilon_t^m \sim \mathcal{N}(0, \sigma^2)$ introduces accumulated model divergence $\Delta_t^m$ along the training process as*

$$\mathbb{E}||\bar{\theta}_{t+1} - \theta_{t+1}^m||^2 \leq \frac{4H(M + 1)\sigma^2 \eta_t^2}{M} \tag{25}$$

*Proof of Lemma 4.1.* By Lemma 3.1, and observing that all $\theta_{t+1}^m$ will be synchronized at the synchronization point as Eq. 7 or Eq. 10, we have

$$\mathbb{E}||\bar{\theta}_r - \theta_r^m||^2 = 0,$$

where $r = H_t \leq \lfloor t/H \rfloor$ represents the last synchronization timestamp until iteration $t$. Thus, we have the following equation by iterating Eq. 14 from $t \to r$,

$$\mathbb{E}||\bar{\theta}_{t+1} - \theta_{t+1}^m||^2 = \underbrace{\mathbb{E}||\bar{\theta}_r - \theta_r^m||^2}_{=0} + \sum_{s=r}^{t} \eta_s^2 \mathbb{E}||\tilde{g}_s - \tilde{g}_s^m||^2$$

$$= \sum_{s=r}^{t} \eta_s^2 \mathrm{Var}(\frac{1}{M} \sum_{k=1}^{M} \epsilon_s^k - \epsilon_s^m)$$

$$= \frac{(M + 1)\sigma^2}{M} \sum_{s=r}^{t} \eta_s^2 \mathbb{E}||\bar{\theta}_{t+1} - \theta_{t+1}^m||^2$$

$$= \frac{(M + 1)\sigma^2}{M} \sum_{s=r}^{t} \eta_s^2 \leq \frac{4H(M + 1)\sigma^2 \eta_t^2}{M},$$

We use $\eta_t \leq \eta_r$ for $t \geq r$ and learning rate decay assumption $\eta_r \leq 2\eta_{r+H}$. Note that different learning rate schedule methods do not influence the order of this bound too much. $\square$

*Proof of Theorem 3.2.* By Equation (24), when $\mu = 0$, and $f$ is convex, we have

$$\mathbb{E}||\bar{\theta}_{t+1} - \theta^*||^2 \leq \mathbb{E}||\bar{\theta}_t - \theta^*||^2 + \eta_t^2 \mathbb{E}||\tilde{g}_t - \nabla F_t||^2$$

$$- \frac{1}{2}\eta_t \mathbb{E}(f(\bar{\theta}_t) - f^*) + \frac{2L\eta_t}{M}\sum_{i=1}^{M}\mathbb{E}||\bar{\theta}_t - \theta_t^i||^2 \tag{26}$$

Rearranging Eq. 26, we have

$$\eta_t \mathbb{E}(f(\bar{\theta}_t) - f^*) \leq 2(\mathbb{E}||\bar{\theta}_t - \theta^*||^2 - \mathbb{E}||\bar{\theta}_{t+1} - \theta^*||^2)$$

$$+ 2\eta_t^2 \mathbb{E}||\tilde{g}_t - \nabla F_t||^2 + \frac{4\eta_t L}{M}\sum_{i=1}^{M}\mathbb{E}||\bar{\theta}_t - \theta_t^i||^2 \tag{27}$$

By summing $t$ from 0 to $T - 1$,

$$\frac{1}{T}\sum_{t=0}^{T-1}\eta_t \mathbb{E}(f(\bar{\theta}_t) - f^*) \leq \frac{2\mathbb{E}||\bar{\theta}_0 - \theta^*||^2}{T} + \frac{2}{T}\sum_{t=0}^{T-1}\eta_t^2 \mathbb{E}||\tilde{g}_t - \nabla F_t||^2$$

$$+ \frac{4L}{MT}\sum_{t=0}^{T-1}\eta_t \sum_{i=1}^{M}\mathbb{E}||\bar{\theta}_t - \theta_t^i||^2. \tag{28}$$

For gradient estimation error from the noise, we have

$$\mathbb{E}||\tilde{g}_t - \nabla F_t||^2 = \mathbb{E}||\frac{1}{M}\sum_{i=1}^{M}g_i(\theta_t^i) + \frac{1}{M}\sum_{i=1}^{M}\epsilon_t^i - \frac{1}{M}\sum_{i=1}^{M}g(\theta_t^i)||^2$$

$$= \mathbb{E}||\frac{1}{M}\sum_{i=1}^{M}g_i(\theta_t^i) - \frac{1}{M}\sum_{i=1}^{M}g(\theta_t^i)||^2 + \mathbb{E}||\frac{1}{M}\sum_{i=1}^{M}\epsilon_t^i||^2$$

$$+ \underbrace{\frac{1}{M}\sum_{i=1}^{M}\mathbb{E}\langle g_i(\theta_t^i) - g(\theta_t^i), \epsilon_t^i\rangle}_{=0} \tag{29}$$

$$= \frac{1}{M^2}\sum_{i=1}^{M}\mathbb{E}||g_i(\theta_t^i) - g(\theta_t^i)||^2 + \frac{1}{M^2}\sum_{i=1}^{M}\mathbb{E}||\epsilon_t^i||^2$$

$$\leq \frac{\sigma_g^2 + \sigma^2}{M}$$

Combining Eq. 29 and Lemma 4.1 into Eq. 28, we have

$$\frac{1}{T}\sum_{t=0}^{T-1}\eta_t \mathbb{E}(f(\bar{\theta}_t) - f^*) \leq \frac{2\mathbb{E}||\bar{\theta}_0 - \theta^*||^2}{T} + \frac{2(\sigma_g^2 + \sigma^2)}{TM}\sum_{t=0}^{T-1}\eta_t^2$$

$$+ \frac{4L\sigma^2(M+1)}{TM}\sum_{t=0}^{T-1}\eta_t \sum_{s=0}^{t-1}\eta_s^2, \tag{30}$$

which completes the proof. $\square$

### E.5 Convergence with noised training with PAFT−Sync.

Here, we use the Martingale Lemma (Lemma 3.3 in (Stich et al., 2018)) to help our proof.

**Lemma E.5.** *Let* $\{a_t\}_{t\geq 0}, a_t \geq 0, \{e_t\}_{t\geq 0}, e_t \geq 0$ *be sequences satisfying*

$$a_{t+1} \leq (1 - \mu\eta_t)a_t - \eta_t e_t A + \eta_t^2 B + \eta_t^3 C, \tag{31}$$

*for* $\eta_t = \frac{4}{\mu(a+t)}$ *and constants* $A > 0, B, C \geq 0, \mu > 0, a > 1$. *Then we have*

$$\frac{A}{S_T}\sum_{t=1}^{T-1}w_t e_t \leq \frac{\mu a^3}{4S_T}a_0 + \frac{2T(T+2a)}{\mu S_T}B + \frac{16T}{\mu^2 S_T}C, \tag{32}$$

*for* $w_t = (a+t)^2$ *and* $S_T \triangleq \sum_{t=0}^{T-1}w_t = \frac{T}{6}(2T^2 + 6aT - 3T + 6a^2 - 6a + 1) \geq \frac{1}{3}T^3$.

**Theorem E.6** (Convergence with noised training with `PAFT-Sync` ( 4.2).)**.** *With object function defined in Eq. 1 satisfying Assumption 3.1, DSGD with* `PAFT` *(Eq. 7 or 10) noise* $\epsilon_t^m \sim \mathcal{N}(0, \sigma^2)$, *we have,*

$$\mathbb{E}f(\hat{\theta}_T) - f^* \leq \frac{\mu a^3}{2S_T}||\theta_0 - \theta^*||^2 + \frac{4T(T+2a)(\sigma_g^2 + \sigma^2)}{\mu M S_T}$$

$$+ \frac{256T}{\mu^2 S_T}\frac{(M+1)}{M}\sigma^2 HL$$

*where* $\hat{\theta}_T = \frac{1}{MS_T}\sum_{m=1}^{M}\sum_{t=0}^{T-1} w_t \theta_t^m$, *for* $w_t = (a+t)^2$ *and* $S_T = \sum_{t=0}^{T-1} w_t \geq \frac{1}{3}T^3$

*Proof of Theorem 4.2.* Using Lemma E.2, Eq. 29, Lemma 4.1 we get

$$\mathbb{E}||\bar{\theta}_{t+1} - \theta^*||^2 \leq (1 - \mu\eta_t)\mathbb{E}||\bar{\theta}_t - \theta^*||^2 + \frac{\sigma_g^2 + \sigma^2}{M}\eta_t^2$$

$$- \frac{1}{2}\eta_t\mathbb{E}(f(\bar{\theta}_t) - f^*) + \frac{8LH\sigma^2(M+1)}{M}\eta_t^3 \tag{33}$$

By Lemma E.5 and the convexity of $f$, rearranging Eq. 33, we have

$$\mathbb{E}f(\hat{\theta}_T) - f^* \leq \frac{\mu a^3}{2S_T}||\theta_0 - \theta^*||^2 + \frac{4T(T+2a)(\sigma_g^2 + \sigma^2)}{\mu M S_T}$$

$$+ \frac{256T}{\mu^2 S_T}\frac{(M+1)}{M}\sigma^2 HL \tag{34}$$

□

# F  MORE EXPERIMENTAL RESULTS

## F.1  ELIMINATE MODEL DIVERGENCE

Fig. 14 shows that the in noised DSGD, the model divergence is accumulated during trainig, thus severely influencing convergence. While the `PAFT` can effectively illuminate the model divergence periodically, thus improving the convergence.

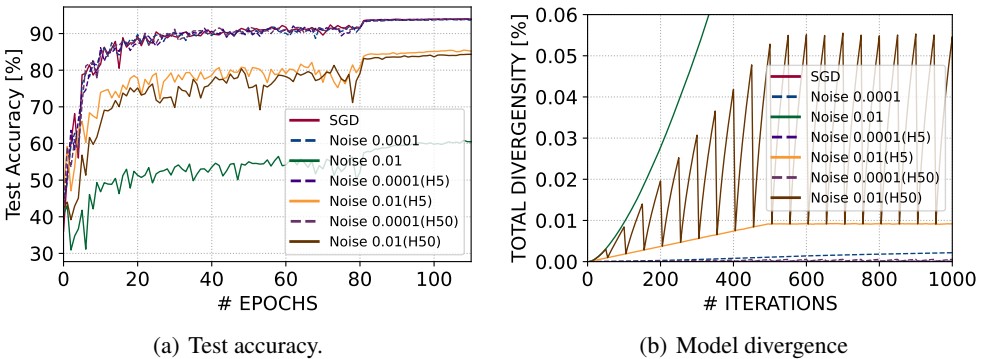

(a) Test accuracy.                    (b) Model divergence

Figure 14: Training ResNet-18 with 4 workers.

## F.2  CONVERGENCE UNDER LARGER NOISE

Fig. 15 and  16 show results of training ResNet-18, ResNet-50 and LLMs with larger noise degrees ($\sigma^2 = 0.1$ or $1.0$). Under the more severe noises, the convergence of LLMs is significantly influenced. And it is more difficult for `PAFT` to mitigate these erros. Nevertheless, such a large noise degree is not common in practice.

## F.3  COMPARING SYNCHRONIZING OPTIMIZER STATES

Fig. 17 provides results of comparing `PAFT` with synchronizing model or all parameters (including optimizer states). The results show that synchronizing all parameters can improve the convergence

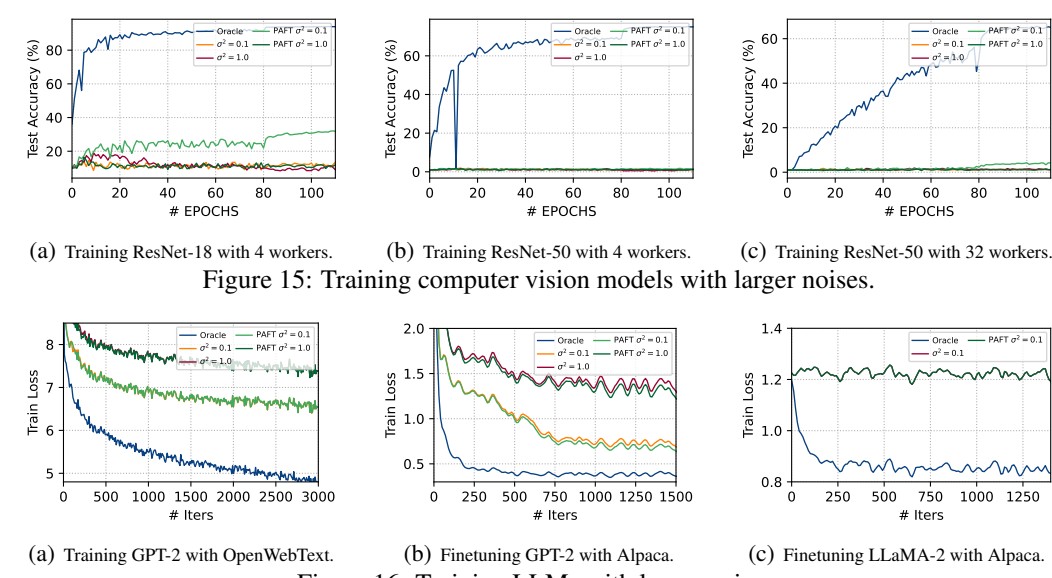

(a) Training ResNet-18 with 4 workers.  (b) Training ResNet-50 with 4 workers.  (c) Training ResNet-50 with 32 workers.

Figure 15: Training computer vision models with larger noises.

(a) Training GPT-2 with OpenWebText.  (b) Finetuning GPT-2 with Alpaca.  (c) Finetuning LLaMA-2 with Alpaca.

Figure 16: Training LLMs with larger noises.

than synchronizing model only. However, the improvement is limited, and the overhead of synchronizing all parameters is much higher than synchronizing model only. Thus, synchronizing model only is more practical in distributed training.

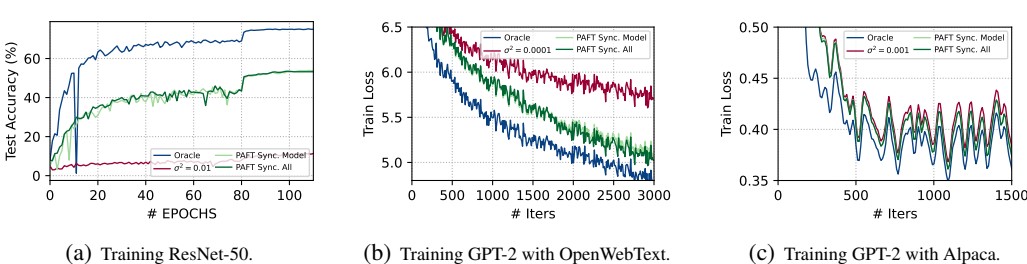

(a) Training ResNet-50.  (b) Training GPT-2 with OpenWebText.  (c) Training GPT-2 with Alpaca.

Figure 17: Comparing PAFT with synchronizing the model or all parameters (including optimizer states). The "Sync. All" denotes synchronizing all parameters, including optimizer states.

