# OpenReview forum: "Identifying and Mitigating Errors in Gradient Aggregation of Distributed Data Parallel Training"
_ICLR.cc/2026/Conference — ICLR 2026 Conference Withdrawn Submission_

### Official Review · Reviewer_Gyeq · 2025-10-27

**Soundness:** 3
**Presentation:** 2
**Contribution:** 1
**Rating:** 2
**Confidence:** 3

**Summary:**

This work investigates silent data corruption (SDC) errors, such as bit corruptions or communication noise, that occur during the gradient aggregation (GA) phase of distributed data-parallel (DDP) training.

1.  The authors mathematically formulate these GA errors as gradient inconsistency. This is modeled as worker-specific additive noise applied to the otherwise correct averaged gradient, such that each worker receives a slightly different gradient. This inconsistency is shown to cause an accumulated model divergence, where the parameters of the DDP replicas diverge from one another, leading to degraded model quality or failed convergence.

2.  To mitigate this divergence, the paper follows the well-known Local SGD/Federated Averaging approapch, proposing a system named PAFT. The primary component, PAFT-Sync correspnds to Local SGD where the local worker gradients are perturbed variations of the true underlying gradient. Analogous to algorithms like Local SGD, PAFT-Sync performs a global averaging of the model parameters across all workers every set number of training iterations. This operation eliminates the accumulated model divergence, resetting it to zero at each synchronization point and ensuring the divergence remains bounded.

3.  The work also proposes PAFT-Dyn, a component designed to minimize the communication overhead associated with frequent synchronization. PAFT-Dyn dynamically adjusts the synchronization frequency based on the profiled error degree. The schedule is determined by the signal-to-noise ratio (SNR), balancing gradient magnitude (signal) against the estimated noise magnitude (noise).

**Strengths:**

1. Formulating gradient inconsistency issues as gradient noise applied to the local worker is an elegant approach that naturally leads to a Local SGD-based solution.
2. The proposed method effectively mitigates the targeted gradient accumulation errors with bearable additional communication and wall-time costs.
3. The PAFT-Dyn method simplifies the hyperparameter tuning process by transitioning from arbitrary intermittent communication to a criterion-based method.

**Weaknesses:**

1. While the paper's formulation of hardware-induced gradient aggregation errors as gradient inconsistency is a novel diagnostic contribution, the subsequent algorithmic and theoretical solutions are largely applications of well-established principles from the Local SGD and Federated Averaging literature. This limits the work's fundamental novelty from an optimization perspective.  For an inexhaustive set of examples, a) parameter averaging is theoretically analysed in [1] and a large body of follow-up works, b) methods to determine when to communicate the gradient/model (PAFT-Dyn) are explored in works such as [2], c) theoretical analysis on how to handle the optimizer states (present in the appendix) is provided by [3].
2. While the paper provides quantitative data on real-world hardware faults in Appendix D, it does not explicitly derive its simulation parameters (e.g., the noise level) from this quantitative data. This creates a gap between the observed real-world problem and the experimental simulation. Without a clear justification linking the magnitude of the simulated noise to the frequency and impact of the observed hardware faults, it is difficult to assess the practical effectiveness and true impact of the PAFT system in a production environment.
3. The additional overheads in wall-clock time of 10-20% are quite high in practice. However, to draw definitive solutions, additional quantification across model types, the number of workers, and the types of hardware would be necessary.





[1] Stich, et.al; "Local SGD Converges Fast and Communicates Little"

[2] Chen, et.al; "LAG: Lazily Aggregated Gradient for Communication-Efficient Distributed Learning"

[3] Cheng, el.al: "Convergence of Distributed Adaptive Optimization with Local Updates"

**Questions:**

1. Could you establish a correlation between the experimental noise levels employed and the empirical rates of hardware malfunctions?
2. Could you please indicate how the overhead costs associated with the wall-clock time would scale with: a) the model size, and b) the number of workers?
3. Are you aware of any data that indicates how the frequency and magnitude of gradient inconsistency issues generally scale with: a) model size, and b) the number of workers?

---

### Official Review · Reviewer_sJJY · 2025-10-28

**Soundness:** 2
**Presentation:** 2
**Contribution:** 1
**Rating:** 2
**Confidence:** 4

**Summary:**

This paper studies silent data corruption during gradient aggregation in distributed data-parallel training. It models SDC as inconsistent gradients where each worker receives an averaged gradient plus independent Gaussian noise. On this model, the authors analyze divergence/failed convergence and propose PAFT, consisting of (i) PAFT‑Sync: periodic parameter averaging every H steps and (ii) PAFT‑Dyn: asynchronous overlap of parameter averaging with training and dynamic adjustment of H using an SNR-like heuristic.

**Strengths:**

The paper formalizes GA SDC as per‑worker additive noise and links it to accumulated divergence and degraded convergence. And a concrete implementation with asynchronous parameter averaging and dynamic frequency scheduling is described with pseudocode and timelines.

**Weaknesses:**

1. PAFT‑Sync’s central mechanism—averaging parameters every H steps—is essentially periodic model averaging, long studied in Local‑SGD lines of work. PAFT‑Dyn adds dynamic H and overlap, but experiments do not compare against standard strong baselines (Local‑SGD) or robust GA rules (coordinate‑wise median, trimmed‑mean, Krum/Bulyan). Most comparisons are to “Oracle / Noised DSGD / PAFT‑Sync with fixed H”, making it hard to isolate PAFT’s incremental contribution from known periodic averaging techniques.

2. All theory/experiments assume independent, zero‑mean, isotropic Gaussian noise per worker, whereas the paper itself summarizes production failure modes—NVLink/PCIe/link errors, ECC events, GSP issues that are bursty, correlated across parameters or devices, and often biased/structured. The “accidental” noise is simulated by occasionally sampling a larger variance Gaussian every 500 steps, which still misses the practical type of corruption.

3. Directly after Theorem 3.2, the paper claims the third term $T_3$ only converges when setting $\eta_t = 0$.  For common decays (e.g., $\eta_t = c/t$), we have $\sum_s \eta_s^2 = O(1)$ and $\sum_t \eta_t = O(\log T)$, so  $T_3 = O((\log T)/T) \to 0$ without setting the learning rate to zero.  This overstates the inevitability of divergence under noise and weakens the necessity argument for parameter synchronization.

**Questions:**

1. How does PAFT compare (accuracy and overhead) to Local‑SGD / Elastic Averaging SGD and robust GA rules (median, trimmed‑mean, Krum/Bulyan) under the same noise schedules? This seems essential to establish incremental novelty.

2. In Alg. 2, how are p,s chosen for $\sigma_{est}$? Does computing $H_{new}$ require an extra all‑reduce every detection round, and is that cost included in Table 1?

3. For asynchronous overlap (Fig. 4), do workers need two copies of parameters (train vs. to‑be‑replaced)? What is the memory overhead, and how are Adam states kept consistent during swap?

---

### Official Review · Reviewer_dyLc · 2025-10-31

**Soundness:** 3
**Presentation:** 3
**Contribution:** 3
**Rating:** 6
**Confidence:** 3

**Summary:**

This paper addresses the issue of silent data corruption (SDC) in distributed deep learning, which causes inconsistent gradients among worker nodes and degrades overall training performance. To tackle this challenge, the authors propose PAFT, a fault-tolerant distributed training framework that includes two components: PAFT-Sync and PAFT-Dyn. Through mathematical formulations, the paper explains how gradient inconsistency arises and then elaborates on the design and functionality of these two modules. The proposed methods not only mitigate gradient inconsistency effectively but also reduce synchronization costs. The experimental results are well-presented and provide convincing evidence to support the framework’s effectiveness.

**Strengths:**

- The experimental section evaluates models with different architectures and various training methods, demonstrating the versatility and general applicability of the proposed approach.
- The paper provides a rigorous mathematical formulation of the problem, ensuring theoretical soundness and clarity.
- By simulating performance under different levels of noise, the authors further explore the method’s applicability range. In addition, by addressing gradient inconsistency while considering performance efficiency, the study highlights its practical value.

**Weaknesses:**

- This paper is based on the assumption that the synchronization process is not affected by SDC. However, this assumption does not fully align with real-world scenarios, as synchronization is also likely to be impacted when SDC occurs. Therefore, the authors need to provide more discussion and justification to support the validity of this assumption.

- The paper does not clearly specify the parameter sizes of the models used in the experiments. This raises the question of whether the proposed method is only applicable to models within a certain parameter range. It is recommended that the authors indicate the model parameter sizes to enhance clarity and reproducibility.

- In Equation (7), the origin of the 1% value is unclear. The authors are encouraged to clarify how this value was determined—whether it was empirically set or derived from experimental results—to help readers better understand the rationale behind this choice.

**Questions:**

In Equation (7), is the 1% parameter empirically chosen or derived from experimental results?

---

### Official Review · Reviewer_ELu7 · 2025-11-09

**Soundness:** 2
**Presentation:** 2
**Contribution:** 1
**Rating:** 2
**Confidence:** 5

**Summary:**

The submission claims to introduce a technique for capturing and mitigating the errors in gradient accumulation in distributed SGD due to silent data corruption (SDC).

To this end, the submission presents the "PAFT" algorithm. The main feature of this algorithm is that it periodically synchronizes the models in a way quite similar to the standard local SGD algorithms. In a variant of this algorithm, which is fully synchronous periodic model averaging, in addition to the gradient communication after every gradient update step, the claim is that the algorithm suffers from high communication cost, whose mitigation requires the presentation of a variant where the averaging frequency is determined based on the model divergence measure.

The algorithms are evaluated on ResNet-18 with CIFAR-10 and ResNet-50 with CIFAR-100 for 120 epochs, and GPT-2 with Open WebText for 3,000 iterations. Additionally, pre-trained LLaMA2 and GPT-2 are trained for an epoch on the Alpaca dataset using the Low-Rank Adaptation scheme. In all these experiments, SDC is simulated as white noise.

The submission also includes a convergence discussion of the presented algorithms.

**Strengths:**

The motivation of the approach is straightforward and relevant.

**Weaknesses:**

The idea of this work is poorly conceived. In particular,

+ It is unclear why the discussion did not cite any of the local SGD papers. For example, a comparison with "Don't Use Large Mini-Batches, Use Local SGD, Lin et al. 2018" will be a relevant approach, where for multiple first epochs, distributed SGD is applied, and after that, periodic averaging is performed.

+ The periodic model averaging on top of gradient communication after every computation is an overkill. Can't the generated/simulated SDC vector be sent in the next round added to the gradient, much similar to the error feedback method? See "Elastic Consistency: A Practical Consistency Model for Distributed Stochastic Gradient Descent, Nadiradje et al. 2022". Once it is modeled with error feedback, it then automatically fits the Elastic Consistency framework for analysis.

+ The simulated SDCs do not seem to be "capturing" it. Can the authors elaborate on some real-life cases of SDC that may be modeled as the standard N(0,1) distribution?

+ The convergence results in the main body of the paper do not even include the convex nature of the objective, which is mentioned only in the appendix. Thus, the results statements still need to be completed. It distracts a reader even when reading the derivations.

+ Training ResNet18/50 models for 120 epochs on CIFAR 10/100 data is not standard. A more standard benchmark is training these models for 300 epochs. Is there any specific reason for using 120 epochs only? It looks more like training to subpar accuracy, an area where different training methods behave starkly differently but, after 200 or so epochs, are close to the known best results. Similarly, for other models.

**Questions:**

Could you please clarify the updates you have made since submitting to ICLR 2025? If I am not mistaken, you have made the same mistake of not specifying the nature of convexity in the main body of the paper, even in this submission, as you did then. In any case, please clarify the updates in your submission from previous iterations; whether you could incorporate the suggestions of the reviewers, or justify why you ignored them.

---

### Note · Authors · 2025-11-25

I have read and agree with the venue's withdrawal policy on behalf of myself and my co-authors.